# A Benefit-Cost Analysis of BackTrack, a Multi-Component, Community-Based Intervention for High-Risk Young People in a Rural Australian Setting

**DOI:** 10.3390/ijerph191610273

**Published:** 2022-08-18

**Authors:** Simon Deeming, Kim Edmunds, Alice Knight, Andrew Searles, Anthony P. Shakeshaft, Christopher M. Doran

**Affiliations:** 1Health Research Economics, Hunter Medical Research Institute (HMRI), Newcastle, NSW 2305, Australia; 2Faculty of Medicine and Health Sciences, University of Newcastle, Newcastle, NSW 2300, Australia; 3Sax Institute, Glebe, NSW 2037, Australia; 4National Drug and Alcohol Research Centre (NDARC), University of New South Wales, Syndey, NSW 2502, Australia; 5Cluster for Resilience and Wellbeing, Appleton Institute, Central Queensland University, Brisbane, QLD 4000, Australia

**Keywords:** youth crime, community, intervention, BackTrack, cost-benefit analysis

## Abstract

BackTrack is a multi-component, community-based program designed to build capacity amongst high-risk young people. The aim of this study was to conduct a benefit-cost analysis of BackTrack, which was implemented in Armidale, a rural town in New South Wales, Australia. Costs and benefits were identified, measured and valued in 2016 Australian dollars. Costs were estimated from program financial and administrative records. Benefits were estimated using a pre-post design and conservative economic assumptions. Benefits included education attendance or completion; employment; engagement with health service providers; reduced homelessness; economic productivity; reduced vandalism to local infrastructure; reduced youth crime; reduced engagement with the justice system; and program income generated by participants. The counterfactual baseline was zero educational outcome, based on discussions with BackTrack staff and expert informants. We tested this assumption compared to the assumption that participants had a Year 8 education. There was evidence of significant quantifiable improvements in several outcomes: high school attendance or completion, vocational education attendance or completion, unskilled or vocationally qualified employment and economic productivity. Reduced homelessness, engagement with health services and acquisition of job readiness skills, as well as reduced local infrastructure vandalism and reduced crime were further quantifiable improvements. The net social benefit of BackTrack was estimated at $3,267,967 with a benefit-cost ratio of 2.03, meaning that every dollar invested in BackTrack would return $2.03 in benefits. BackTrack represents a viable funding option for a government interested in addressing the needs of high-risk young people.

## 1. Background

BackTrack is a community-based program designed to build capacity amongst high-risk young people who typically have multiple and complex needs [1,2,3]. The program provides holistic, flexible and long-term support for young people who are having a hard time. It is a multifaceted program that aims to: connect and engage with young people in need; provide individualised care and wellbeing; organise diversionary activities; support personal development; provide learning, training, work experience and employment opportunities; and help young people to contribute and connect with their community (http://www.backtrack.org.au, accessed on 22 July 2022). More specifically, BackTrack’s activities can be organised into five standardised core program components: (i) engagement, to optimise participation in the program; (ii) case management, to provide individual life skills management; (iii) diversionary activities, to reduce exposure to police and the judicial system and provide recreational opportunities; (iv) personal development, identity and team identity, to improve health and wellbeing; and (v) training and skill development, to increase opportunities for active participation in education or training that increases the likelihood of employment [1]. Details on the BackTrack program are available elsewhere [1,2].

For the majority of program participants, the key risk factors are criminal activity, school absence, lack of employment, high-risk of mental health issues (including suicide ideation), alcohol and other drug problems and lack of engagement with and utilisation of the health system [2]. BackTrack offers participants the opportunity to reduce their vulnerability to these risk factors and re-engage with their community. Given it is an intensive program for a relatively small number of high-risk young people, the extent to which it represents value for money is an important consideration that can be addressed through economic analysis. The objective of economic evaluation is to identify, measure and value what society forgoes when it funds an intervention (the opportunity cost) and what it gains (the benefit) [4]. Economic evaluation thus provides an important evidence base to inform policy makers’ decisions about the allocation of society’s scarce resources. A recent systematic review of interventions for high-risk young people found no existing published evidence of economic evaluations for multi-component, community-based programs like BackTrack [5]. Consequently, the aim of this study was to address this lack of evidence by conducting a retrospective economic analysis of BackTrack.

## 2. Methods

### 2.1. Economic Framework

Benefit-cost analysis (BCA) is the appropriate economic analysis to capture the multiple social, health and economic outcomes arising from social programs like BackTrack [4]. BCA compares the benefit of an intervention with its costs. The results are presented as both a net present value (NPV) and a benefit-cost ratio (BCR) (benefits/costs), where a positive ratio indicates that the benefits have outweighed the costs. This retrospective BCA conformed to the New South Wales Government’s Treasury guidelines [6] and the Consolidated Health Economic Evaluation Reporting Standards (CHEERS) Statement [7].

#### 2.1.1. Setting, Participants, Perspective, Timing

The BackTrack program was implemented in Armidale, a rural town in New South Wales (NSW). Armidale has a population of close to 24,000 with a land area of 7818 square km, representing a population density of 3.77 persons per square km [8]. The town is classified as an area of low socio-economic disadvantage with an unemployment rate of 7.3% in 2016 [8]. The referral process and general characteristics of participants are provided elsewhere [2], and 34 participants were eligible for inclusion in this BCA. 

The BCA adopted a societal perspective and covers a nine-year period, comprising of five years of program operation (March 2012–February 2017) and four years of additional projected benefits (March 2017–February 2021). All estimated cost and benefit values were converted to 2016 Australian dollars using the Australian Gross Domestic Product Implicit Price Deflator [9]. Both costs and benefits were discounted at an annual rate of 7%, with sensitivity at 3% and 5% [6].

#### 2.1.2. Baseline Specification

For both the individual participants and their communities, the extent of change from pre-intervention to post-intervention was determined by a counterfactual analysis [10,11]. The counterfactual baseline for the analysis reflected pre-intervention measures and assumed projections in the absence of the program. The basis for the projections varies in line with specifications for each cost and benefit component (see below), while retaining consistency with respect to the economic theoretical framework, the study boundary, the investment and the participants in receipt of the program. However, the central projection underpinning all the variable costs and benefits comprises the number of participants eligible for this study (*n* = 34) and their level of participation with the program, school, further education and employment. For the specified program investment period, program data accurately recorded participant status. For the post-trial projected period, a decay rate was applied per annum to reflect that in the absence of the program’s support, participants may not be able to sustain their involvement in education/employment and as a consequence, the beneficial gains (e.g., in crime and health) may not be sustained. The analysis assumed a conservative decay rate of 50%, implying that half of the participants engaged in the preceding year in school, employment or other activities would drop out in the following year. This conservative assumption is tested in the scenario analysis.

The productivity benefits arising from education and employment used the human capital approach to value the progression of a participant from one status to another i.e., from school attendance to high school completion, from unemployment to employed. The human capital approach is an accepted method in economic analysis to estimate the social benefit arising from this progression. The projections in the absence of the program reflect the average productivity of males in Armidale, assuming that their income follows the average career trajectory for men with equivalent qualifications. The projections with the program apply the same approach, but were derived from males in Armidale with higher levels of qualifications/employment status. The projected productivity benefits incorporate proportional labour force participation and unemployment assumptions. In line with the conservative approach, participants were assumed to retire at 60 years, at which point no further benefits are realised. The basis for the specific crime projections is explained in detail below and in Appendix A.

#### 2.1.3. Program Attribution

It is possible that participants may have realised productive outcomes without the provision of the BackTrack program, due to their innate capabilities. Previous research on the program suggests that the productive outcomes for participants dropping out of the program and for the community from which the participants are drawn are extremely low [1,2]. Similarly, evidence from the educational economics literature suggests the beneficial uplift from education is higher for lower socio-economic groups, where innate talent may be less able to be activated [12]. To account for this innate talent, the productivity benefit estimated that each respective employment/education outcome is adjusted by 5%. This is an estimate based on the insights from the program managers and the research team. Further, decay rates are applied to appropriate assumptions to reflect waning benefits beyond the investment period. Individual lifetime benefits from education or employment are derived for projected careers from ages 18 to 60 years. 

#### 2.1.4. Identification of Costs and Benefits

A systematic literature review of economic evaluations of programs for high-risk young people was conducted to inform the potential costs and benefits arising from BackTrack [5]. In accordance with the NSW Government’s Treasury guidelines, the range of possible impacts (costs or benefits) include both primary and secondary (flow-on) markets [6]. Direct costs in the primary market refer to costs such as the purchase of equipment, operating costs and staff wages. Direct benefits include the additional economic productivity generated by the participants through their productive life, reductions in homelessness, improved health and reductions in health system utilisation. Indirect effects include reductions in infrastructure vandalism costs, policing and justice system costs. Intangible impacts refer to program-related improvements such as a sense of belonging, mental health, family patterns and perceptions of public safety. 

### 2.2. Measurement and Valuation of Costs 

The costing model was developed from program financial and administrative records alongside discussions with the program manager and staff. Program inputs were included for the five core components of the program (engagement; case management; diversionary activities; personal development; and training and skill development) [1], and comprised of opportunity costs for infrastructure and equipment, labour and non-labour inputs. There is also an opportunity cost for the time that participants attended the program. Participants could have utilised this time for employment, education or leisure purposes. The valuation of this cost considers that upon entry to the program, participants were not employed and rarely attended education. 

To account for the costs associated with greater participant engagement with the health system, a corresponding cost for the provision of four consultations per annum with each program participant reporting higher health service engagement was incorporated. The costs corresponded to the number of program participants for whom a health benefit was conferred [2]. The cost of tax transfers, $1.25 for every dollar of tax revenue raised, was also included to account for the Australian Taxation Office’s administration of taxes [6,13].

### 2.3. Measurement and Valuation of Benefits

#### 2.3.1. Job Readiness, Literacy, Numeracy

BackTrack provides education to improve participants’ basic literacy and numeracy skills. It is assumed that attendance at BackTrack for a minimum period of 12 months collectively is sufficient for participants to realise this benefit, the lowest education/employment outcome for which a benefit has been acknowledged. Therefore, all participants who attend the BackTrack classroom for 12 months receive the job readiness, literacy and numeracy skills benefit, valued at $4677 (AUD2016) as a one-off value, irrespective of whether they went on to realise further employment or education achievements [14]. 

#### 2.3.2. High School Education

The high school education outcomes from the program fall into two categories: participants recommencing high school attendance, or participants completing high school stages. The additional annual cost of a full-time secondary student (classroom) is $21,149 and $622 in administration, a total of $21,771 (AUD2016) [15]. The human capital method was used to calculate productivity gains for program participants completing different levels of education whilst at BackTrack. This method assumes that employee income reflects the market value of productivity within the economy. The total personal income profile for educational outcomes was derived from census data for males who are usually residing in Armidale [16]. It was assumed that 80% of the participants attending school will realise their respective completion stage. The NPV of the improvement in productivity was also modified for the innate talent assumption (5%), and a proportional attribution to the BackTrack program (50%) as opposed to education alone.

#### 2.3.3. Vocational Further Education

The same human capital method was applied to the benefit of vocational education. The additional cost of a Certificate I/II Vocational Education and Training (VET) qualification (government and private) is $7763 (AUD2016) [17]. As per secondary education outcomes, these cross-sectional data were used to estimate the income profile for males with Year 8 high school or below and vocational education [16]. The productivity gains for a participant progressing from Year 8 to Certificate I/II vocational education are assumed to reflect the difference in income profiles for these two cohorts. The NPV for each participant progressing to Certificate I/II reflects the discounted gains in personal income from age 15 for VET to age 60. As for other educational outcomes, it is assumed that 80% of the participants attending VET will realise their respective completion. The NPV is also modified for the innate talent assumption (5%), and a proportional attribution to the BackTrack program (50%) as opposed to education alone.

#### 2.3.4. Health System Engagement

Upon entry to the program, participants were surveyed to provide a baseline measure of health service utilization [2]. Analysis of the survey results at the six-month follow-up found significant improvements in the engagement with a health professional component. A benefit has been attributed to BackTrack for this improvement because health literacy and health system engagement represent an explicit component of the program. To maintain the conservative approach, benefits were only allocated to the participants demonstrating improvement. In the absence of economic values for health service engagement, the benefit has been valued with reference to the health service engagement associated with improved health literacy. A systematic review found that the difference in annual health care costs for patients with poor health literacy, compared to average, varied from US $143 (AUD207)–US $7798 (AUD11,307) per person per annum [18]. A conservative estimate of $2541 (AUD2016) per annum was assumed for the participants demonstrating an improvement in this outcome. The health benefit was conservatively assumed to commence after 12 months of participation in the program investment period and be sustained for ten years after the participant exits the program.

#### 2.3.5. Homelessness

In June 2015, a residential site was made operational for BackTrack participants without secure accommodation. It was comprised of six permanent beds and two emergency beds. It was assumed that permanent would equate to three years, the usual time a participant would stay in BackTrack allowing him/her to develop sufficient life skills and reach an age where it would be possible to secure other accommodation and become economically productive to sustain that situation. The benefit for one year was valued at $14,961 (AUD2016) per participant [14]. In addition, program managers estimated that, on average over a period of three weeks, several participants used the two emergency beds. As the benefit can only be claimed for a successful transition from homelessness to temporary accommodation, it was assumed that only ten per cent of those utilising the emergency accommodation, subsequently transitioned to more permanent accommodation with a relative or an alternative resolution of their homelessness. The remainder continued to cycle in and out of homelessness and realised no additional benefit. The additional transition generated a value of $24,967 (AUD2016) per participant [14].

#### 2.3.6. Economic Productivity—Workforce Participation

The economic productivity outcomes from the program fell into two categories: unskilled employment and vocational employment. The human capital method was utilised to estimate the productivity gain arising from the progression of male youth from a zero education baseline to unskilled or vocational employment. Adoption of the human capital method was consistent with the labour costs of program delivery. Median annual income was estimated using the count data for total personal income for ten-year age cohorts. The NPV for each participant was assumed to be realised after three years in the program. The productivity gains for a participant progressing, for example, from a zero education baseline to unskilled employment was assumed to reflect the difference in income profiles for these two cohorts. The NPV for each participant progressing to full-time unskilled employment or vocational employment reflected the discounted gains in personal income from age 18 for both unskilled employment and vocational employment to age 60. Employment income for part-time work was equivalent to 52% of full-time income. The NPV was modified for the innate talent assumption (5%), and a proportional attribution to the BackTrack program (50%) as opposed to vocational education alone.

#### 2.3.7. Local Infrastructure

The Armidale Dumaresq Council Crime Prevention Strategy 2014–2018 identified that, in 2013, 50% of crime perpetrated by males (responsible for 77% of all crime) was carried out by 10–19-year-olds and 41% by 10–17-year-olds, the age cohorts supported by the BackTrack program [19]. Many of the most frequently committed crimes, such as break-and-enter, retail theft, and malicious damage represent costs to local government in reparations to address physical vandalism and graffiti. The Armidale Dumaresq Graffitti Management Plan 2010 recorded a decline in annual costs to $37,000 in 2009 from peaks of $150,000 in previous years [20]. The conservative approach used in this study precluded the attribution of this decline to the program. However, to acknowledge the likelihood that the program contributed to the decline in council costs, and evidence of a statistical reduction in crime (see below), a reduction of 25% was attributed to the program from the average annual cost ($102,019 AUD2016) of vandalism to local infrastructure. This equated to a value of $25,087 (AUD2016) per annum and was continued during the post-intervention period.

#### 2.3.8. Crime

Criminal incident data related to Armidale were obtained from the NSW Bureau of Crime Statistics and Research (BOCSAR) for males aged 15 to 18 between 1 January 1999 and 31 December 2017. Segmented generalised linear models were used to model count time series data pre- and post-intervention implementation. The data suggested a statistically significant reduction trend after the respective intervention dates (see Appendix A). It was estimated that there was a reduction in the pre-intervention trend of −0.924% per month, compounded monthly. Using estimates from Byrne et al. [21] and Deeming & Kypri [22], the mean weighted cost of crime committed by this population was estimated at $4689 (AUD2016). This value was adjusted for the probability that a crime may or not be reported to the police, noting that crimes reported to the police are more expensive given the higher probability of subsequent engagement with formal criminal justice processes [21]. The attribution to BackTrack for this reduction in crime was 75%.

#### 2.3.9. Program Income

The Learning and Skills component of the BackTrack program was comprised of several work ready activities that generate income. For example: welding and the sale of metalwork sculptures or farm gates; farm work, such as fencing or firewood collection; and dog handling exhibitions. This program income is acknowledged as a benefit reflecting the social value of these services to the community.

### 2.4. Discrete Choice Experiment

A Discrete Choice Experiment (DCE) was used to quantify, in monetary terms, the value that households in the BackTrack community placed on improving outcomes for young people with multiple and complex needs [23]. DCEs involved the use of a survey to systematically quantify individuals’ preferences in relation to a number of characteristics (attributes) of a program [24,25,26,27]. The value placed by a respondent on a given attribute can be quantitatively measured against another attribute. DCEs thus provided a method to assess the community’s perception of a program whilst also deriving a monetary estimate of the program’s value for BCA.

### 2.5. Sensitivity Analysis

The nature of BackTrack, its inherent flexibility and the number of participants passing in and out of the program, implies that there is limited information to inform the probabilities of any given cost or benefit. Consequently, it is not possible to conduct uncertainty analysis, such as bootstrapping, around the parameters included in the analysis. This limitation has been addressed by adopting a rational, but conservative, approach to all assumptions, and by conducting sensitivity analyses. Three types of sensitivity analysis were conducted: (i) parameter; (ii) discount rate; and (iii) deadweight loss (DWL). A multivariate parameter scenario analysis varied the assumptions associated with participant engagement decay rate, education completion rates, innate talent assumption, crime and economic productivity. These scenarios were based on a worst to best case analysis. Consistent with the NSW Government’s Treasury guidelines [6], the base case analysis uses a 7% discount rate for costs and benefits, which was varied to reflect 3%, 5% and 10% discount rates. DWL accounts for the fact that taxes distort consumption choices away from things that are taxed to those which are not taxed (or are lightly taxed). This has an adverse welfare effect, additional to the loss of welfare resulting directly from the loss of money taken in the form of tax. We used 27.5% as the DWL from the taxation revenue raised to provide the public investment in BackTrack [13].

## 3. Results

### 3.1. Counterfactual Baseline

Although the agreed counterfactual baseline for the population of high-risk youth in this BCA was zero educational outcome, we tested this assumption compared to the assumption that participants had a Year 8 education.

### 3.2. Costs and Benefits

Table 1 provides a summary of the costs and benefits of the BackTrack program over the nine-year period commencing in March 2012, with all units valued in 2016 Australian dollars. A year-by-year assessment is provided in Appendix A.

The costs relate to the additional resources that were required to deliver BackTrack. Labour costs account for the majority of the operating costs. The costs included in the post-intervention period capture additional demands on the health service, arising from greater participant engagement with the health system.

The economic benefits associated with employment accounted for over 75% of the direct benefits ($4.95 million). Reductions in crime resulted in an estimated benefit to society of $1.23 million, with higher levels of education accounting for $1.07 million.

### 3.3. Estimating the Benefit-Cost

Table 2 shows the net economic benefit of the BackTrack intervention. The NPV for all effects is estimated at $3,267,967 with a BCR 2.03, meaning that every dollar invested in BackTrack would return $2.03 in benefits, including improved productivity, cost savings from crime, reduced homelessness and reduced health services utilisation.

In the BackTrack DCE, participants were asked to choose between BackTrack and a policing alternative, with different costs and efficacies. The full methods and results of the DCE are published elsewhere [23]. The results of the DCE lend support for community preference for community-based interventions like BackTrack compared to a greater police presence, and valued the intervention at (AUD) $150 per household. The BCR based solely on the DCE or community willingness-to-pay was 2.14.

### 3.4. Sensitivity Analysis

#### 3.4.1. Parameter Variation

As summarised in Table 3, performing multivariate sensitivity analysis on key parameters still resulted in a BCR of 1.13 in the worst-case scenario and 2.17 in the best-case scenario.

#### 3.4.2. Discount Rate Variation

Consistent with the NSW Government’s Treasury guidelines [6], the base case analysis uses a 7% discount rate for costs and benefits. Table 4 provides a sensitivity analysis of results to variations in the discount rate. A lower discount rate improves the return on investment. Even when discounted into the future at the upper rate of 10%, the program still achieved positive returns (BCR 1.44).

#### 3.4.3. DWL

The inclusion of a 27.5% DWL from the taxation revenue raised to provide the public investment in BackTrack contributed $236,771 in costs, reducing the net benefit to $3,053,950 or a BCR of 1.90.

### 3.5. Intangible Benefits

In addition to learning formal skills (detailed in Section 2.3), participation in BackTrack resulted in benefits that are difficult to value. For example, a number of respondents mentioned that their involvement in BackTrack gave them a sense of identity and belonging within the community, which increased their self-esteem and self-respect. They expressed pride in wearing the BackTrack shirt because people in the community acknowledge them and make them feel valued. Similarly, while there were no data to support the economic impact of BackTrack on better physical and mental health, this population typically has high levels of substance abuse and poor mental health [3,28,29,30]. Reduced exposure to drugs, alcohol and violence has a positive intergenerational impact [31] and the potential to generate considerable downstream benefits in improved educational attainment, employment, reduced exposure to the justice system and reduced health service utilisation [3,32,33]. Given the high risk of poor mental health and community alienation amongst this population, there is a high risk of suicide [28,29]. The reduced incidence of suicide from improved mental health represents a benefit to the individual, family and wider society [28,29,33]. Other intangible benefits of BackTrack are those that impact on the community or broader society. For example, the value that arises from workforce engagement, such as reduced job search initiatives and the administration costs of welfare. In addition, antisocial behaviour and crimes such as vandalism and graffiti have been shown to have significant negative impact on property prices, largely through their ability to motivate fear within the community where they are interpreted as signs of instability, disorder and deterioration [34].

## 4. Discussion

At the time of writing, the authors were unaware of any economic evaluations of multi-component interventions for high-risk young people [5]. While there is an awareness of the value of interventions that support such populations to move away from crime, unemployment, substance abuse and unhealthy lifestyles, most interventions are limited to focusing on one or two risk factors. Among those more limited programs, there have been few economic evaluations [2,5,35,36,37].

For this BCA, the effects of BackTrack were estimated using a pre-post design and conservative economic assumptions that allowed participant and community outcomes to be compared to a counterfactual (i.e., what their outcomes would have been in the absence of BackTrack). A broad societal perspective was adopted to produce a comprehensive evaluation to better support informed decision making. Both primary and secondary sources were used to measure and value the costs and benefits of BackTrack over a nine-year time frame. Our analysis also explored the sensitivity of estimates to alternative assumptions about the counterfactual, discount rates and inclusion of DWL. Adoption of government evaluation guidelines also enhanced the methodological rigor and practical application of this analysis.

BackTrack generated a positive return on investment, despite the conservative approach of the analysis and the fact that it is a social program with benefits that tend to have a longer-term impact. When interpreting the results of this BCA, several limitations should be considered. First, the requirement to translate costs and benefits into comparable dollar values naturally favours impacts that can be readily quantified, such as operational costs. This is a recognised concern in BCA because the major costs are often readily quantifiable and founded in the initial stages of a project, whereas the benefits are more likely to be difficult to quantify and typically extend into an uncertain future. Unquantifiable costs or benefits are sometimes referred to as intangibles. The NSW BCA guidelines [6] note that these “will often be very important in public sector projects, and their identification is vital to the process of economic appraisal.” Previous research with BackTrack shows that this consideration is highly applicable to this BCA [1,2]. Second, not all valuation of benefits involved a straightforward quantification. For example, there are limitations with the methodology adopted to value health engagement, given the lack of a control group and the fact that health system engagement is only one aspect of health literacy, meaning a conservative value for health literacy was included. Third, given the flexible nature of the program, the length of follow up does not necessarily reflect exposure to the program, which means that many of the effects included are restricted to the short-run effects available at the time of analysis. Finally, some of the data collected were incomplete (unanswered questions) or had simply not been collected (unfinished follow up). Working with high-risk and marginalised youth exacerbates the difficulty of data collection. Improved data collection over a longer time period would improve the reliability and robustness of longer-term benefits.

## 5. Conclusions

Programs that implement the same standardised components as BackTrack represent a viable funding option for governments interested in addressing the multiple needs of high-risk young people. The flexible implementation options of such standardised programs [1], along with their numerous benefits (as demonstrated in this study) and their strong community support [23], make them attractive from the viewpoint of scalability, translation and sustainability.

## Figures and Tables

**Table 1 ijerph-19-10273-t001:** Summary of costs and benefits, intervention, post-intervention and total.

	Intervention Period	Post-Intervention Period	Total
Costs			
Infrastructure & Equipment	$227,498	$0	$227,498
Operating costs (Labour)	$2,564,950	$0	$2,564,950
Operating costs (Non-labour)	$870,506	$0	$870,506
Additional health service costs	$28,272	$7312	$35,584
Administration of tax transfers	$10,762	$0	$10,762
Total costs (discounted)	$3,180,215	$4836	$3,185,040
Benefits			
Program Income	$306,174	$0	$306,174
Education/Training—job training, literacy & numeracy skills	$135,627	$35,076	$170,703
Education/Training—High school	$594,949	$286,978	$881,928
Education/Training—Vocational further education	$177,464	$44,366	$21,830
Physical health—Engagement with health services	$164,302	$44,010	$208,312
Homelessness & housing	$139,763	$0	$139,763
Employment—Increased productivity	$3,701,124	$1,251,462	$4,952,586
Local Government: Infrastructure vandalism (savings)	$168,331	$0	$168,331
Crime	$1,140,035	$88,332	$1,228,367
Total benefits (discounted)	$5,300,467	$1,152,540	$6,453,007

NB: Intervention period (12 March–16 June), Post-intervention period (16 July–21 February).

**Table 2 ijerph-19-10273-t002:** Economic benefit of the BackTrack program.

	Intervention Period	Post-Intervention Period	Total
NPV costs	$3,185,040	$0	$3,185,040
NPV benefits	$5,300,467	$1,152,540	$6,453,007
Net social benefit	$2,120,262	$1,147,705	$3,267,967
BCR			2.03

NB: Intervention period (12 March–16 June), Post-intervention period (16 July–21 February).

**Table 3 ijerph-19-10273-t003:** Sensitivity analysis using variations in key parameters.

Scenario	Worst	Central	Best
Result			
Benefit-Cost Ratio	1.13	2.03	2.17
Scenario parameters			
Participant engagement decay rate (pa)	75%	50%	25%
High school/Vocational education: Completion rate	70%	80%	90%
High school/Vocational education: Attribution to BackTrack	33%	50%	66%
Innate talent assumption	5%	5%	0%
Vandalism: Attribution to Backtrack (reduction)	17%	33%	50%
Crime: Attribution to BackTrack	33%	50%	66%
Economic productivity: Baseline assumption i.e., without BackTrack	Equivalent to Year 8 high school achievers	Equivalent to no high school outcomes	Equivalent to no high school outcomes

**Table 4 ijerph-19-10273-t004:** Sensitivity analysis using variations in discount rates.

	Sensitivity Analysis 1 (3%)	Sensitivity Analysis 2 (5%)	Base Case (7%)	Sensitivity Analysis 3 (10%)
NPV costs	$3,474,025	$3,323,731	$3,185,040	$2,996,148
NPV benefits	$13,506,061	$9,032,351	$6,453,007	$4,312,190
Net social benefit	$10,032,036	$5,708,620	$3,267,967	$1,316,041
BCA	3.89	2.72	2.03	1.44

## Data Availability

No restrictions apply to the availability of these data and are available from the corresponding author.

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
