# Peer review of "A Benefit-Cost Analysis of BackTrack, a Multi-Component, Community-Based Intervention for High-Risk Young People in a Rural Australian Setting"

_ijerph, 2022, doi:10.3390/ijerph191610273_

Round 1

Reviewer 1 Report

The manuscript ‘A benefit-cost analysis of BackTrack, a multi-component, community-based intervention for high-risk young people in a rural Australian setting’ presents a costs-benefit analysis of a re-education program for young citizens, describing the different fields of intervention and separately evaluating their economic impact on the society, finally giving an estimate of the return on investment.

The study is of high interest, although there are some aspects that in my opinion should be elaborated in the manuscript (see comments below). However, I think it is not compliant with this journal: as regards the environment, in this case it is only addressed as the ‘social’ environment, with no connection to the ‘natural’ environment; with relevance to public health, it only represents a small part (and a secondary one) in the overall context of the study. Therefore, I think that this manuscript should be re-addressed to a different journal, with a more specific focus towards economics and social behavior.

In the following, I propose some suggestion to improve the manuscript:

MAJOR COMMENTS:

1) Introduction: the BackTrack system is hereby described, but only in terms of aims and fields of intervention, I suggest to add a brief explanation of what the program consists of and how it is in facts carried out, which is missing in the manuscript.

2) Line 90: how are these projections developed?

3) Line 99: how is the 5% correction dimensioned? If it is from literature, a reference should be added.

4) Although, as authors state in 2.6, it is not possible no conduct a complete sensitivity analysis, it would at least be necessary to provide multiple scenarios (e.g. worst case, expected average, best case) referred to all costs and benefits (in addition to the presented variations relevant to different discount rates), by for example considering the lower and upper estimates of costs and benefits found in literature.

5) As the research involves very personal and sensitive aspects of the subjects' life, I think that an ethical assessment by a competent committee should be included.

MINOR COMMENTS:

Line 65: probably a typo, maybe it’s ‘found no evidence’.

Line 78: add some information relevant to the town of Armidale, such as population, land-use, socio-economic characteristics, maybe compared with a baseline (region, nationwide, or against a more urbanized area).

Section 3.1 should be in the methods.

Line 286: this sentence results a little confusing due to the repeated use of the word ‘costs’, maybe it should be rephrased.

Reviewer 2 Report

This article provides a cost-benefit analysis of the BackTrack program, which targets youth at high risk of harm. 

The Introduction could do with a bit of work before publication. I wonder if, instead of just outlining the components of BackTrack, you can reference a publication which provides more information about the program itself. I think the Intro would be more suited to include further information about why an economic evaluation of such a program is important. Why is this particular population costly to society and how can programs targeting this group reduce costs? Can actual values be placed on these arguments, supported by existing literature?  It would be great to have more than 3 studies cited in the Introduction so that the importance of this study is fully explained. 

Line 55 - most common risk factors for what?

Line 65 - remove the word 'no' before evidence (I assume). 

Lines 64-65 - what makes economic evaluation important? Does the lack of research exploring economic evaluations in this area perhaps suggest that such evaluations are perhaps not important? I think you need to provide better justification as to why economic evaluations in this space are important. 

Appropriate economic analysis chosen based on study aims. It would be good to have a little more info about the participant sample.  How many people participated in this program and over what time period? It is also not clear to me what the pre- and post-intervention time periods are. 

It was mentioned in the Intro that there was no literature which used economic evaluations in this area. However, in the Methods, a systematic literature review is cited as contributing to the costs and benefits chosen for this paper. What did the review indicate, assuming there was little to no literature to review?

All measures are fairly well explained, and the % accountable for different outcome measures as a result of the BackTrack seem reasonable, if not a little overestimated? I wonder how these accountability figures were agreed upon or if sensitivity analyses were performed (and could be reported, perhaps in supplementary tables) which adjusted for the % in accountability? I understand that this is explained somewhat in the Sensitivity Analysis paragraph, but this paragraph also highlights some of the key limitations of this analysis which perhaps reduce the interpretability and generalisability of key results from this study. I particularly feel as though the fluid nature of participation in this program make it difficult to properly evaluate from an economic perspective. 

I am unsure of the meaning behind lines 245-247. Can you please clarify what is meant here?

Error in line 309 - 'Th'.

Given the lack of clear data on the matter, I wonder if the intangible benefits section is perhaps over-reaching in interpretation of results. This is particularly true for the section focused on mental health and suicide. I have concerns that the authors are trying to link the program to increased mental health, with no evidence to support this claim. Consider removing this section. 

For the Discussion, it would be good to see some information about how the various study limitations could be addressed. A majority of the Discussion is dedicated to limitations, instead of highlighting the importance of the study and context in the wider literature. 

Reviewer 3 Report

The authors report on a cost-benefit analysis of "BackTrack", a community-based intervention program designed to reduce negative outcomes (such as crime, unemployment, not finishing school) among youth in rural town in New South Wales, Australia. The authors compare the benefit of the intervention against intervention costs and report the net present value and benefit-cost ratio of the "BackTrack" intervention program. The authors concluded that the program represented a viable funding option for governments interested in addressing the needs of high-risk young people.

Major concerns:

1. My major concern with this study is how intervention costs were assessed and whether measures of intervention effectiveness were considered in the cost-benefit analysis. In particular, was an evaluation of the BackTrack program conducted (using either a randomized controlled design or some type of observational design) to assess the overall effects of the intervention? In other words, what percentage of youth who participated had beneficial outcomes compared to a "usual care" approach or no intervention? Surely not all youth who participated would have experienced beneficial outcomes? How are these "costs" considered in the analysis (i.e., if an intervention is delivered but a large percentage of youth do not benefit from the program, then societal "benefits" would be expected to be lower while there were costs of the intervention that did not achieve the stated objectives of the program). Is this type of information considered in the analysis?

Indeed, the authors only state in section 2.1.2 that the "counterfactual baseline for the analysis reflected pre-intervention measures and assumed  projections in the absence of the program." How realistic are such assumed projections in absence of rigorous intervention evaluation via a randomized design? (I recognize that a randomized design may not have been possible, however, the point still stands that assuming benefits of the program without considering program efficacy seems somewhat odd). There needs to be some recognition of program effectiveness and how that would be expected to affect the results of the cost-benefit analysis reported.

2. Page 1: clarify what is meant by high-risk young people with "multiple and complex needs." I.e., provide specific examples of what these multiple and complex needs are. It isn't clear based on the information provided the types of youth who would be considered eligible for the BackTrack program and what proportion of youth in rural Australian areas who might be eligible for the program. This would clarify the information in section 2.1.1 (page 2) as well.

3. Page 3, section 2.3.1: "It is assumed that attendance at BackTrack for a minimum period of 12 months collectively is sufficient for participants to realise this benefit." How can these benefits be known without sufficient evaluation data (e.g., from an RCT or at least an observational comparative study) to know what the "success" rates of the program are? E.g., what proportion of participants will have improved outcomes; what proportion of participants might drop out of the program? Are these considerations built into the cost-benefit analysis? What proportion of participants actually complete 12 months of the program? If this isn't considered for the BCA, how does this then influence the estimated benefit-cost ratio? What other assumptions are built in? Do such factors influence calculation of costs and benefits?

4. Section 2.3.2: "The additional cost of a full time secondary student (classroom) is $21,149 and $622 in administration, a total of $21,771." Is this the cost per year or cost per student to complete high school -- clarify.

5. Section 2.3.4: Health care costs were provided in US dollars. It would be better to cite these costs in 2016 Australian dollars for ease of interpretation (costs in USD should be inserted parenthetically behind the 2016 Australian amounts).

6. Table 1 - do the operating costs depend on the number of program participants? If there were more participants, would these costs change?

7. Table 1 - Benefits - Do all benefits assume successful program completion by all participants? What if participants only complete a portion of the program? Does this analysis consider this? This is unclear from description in the methods.

8. Page 7: "The NPV for all effects is estimated at $3,267,967 with a BCR 2.03, meaning that every dollar invested in BackTrack would return $2.03 in benefits such as improved productivity, cost savings from crime, reduced homelessness and reduced health utilisation." Does this statement assume all participants complete the program and have beneficial outcomes? This surely isn't the case, which is why evaluation data (ideally from an RCT, but at least from observational data) should be informing the BCA. It is hard to know how realistic these estimates are without considering objective measures of the program's success... a segmented regression on limited outcomes isn't very convincing in my mind... how much of that was actually attributable to program success? Were difference-in-differences properly evaluated? Was there an assessment of existing trends? Were potential declining trends ruled out?

9. Page 9, Discussion: "For this BCA, the effects of BackTrack were estimated using statistical techniques" -- It's unclear what statistical techniques were really used. Many seem to be "back of the envelope" calculations comparing the BackTrack program to a hypothetical counterfactual.

Minor comments:

1. Ensure consistent hyphenation throughout, e.g., line 22 of the abstract "benefit cost analysis" should be "benefit-cost analysis". Line 41 (page 1): "high risk young people" should be "high-risk young people".

2. Lines 217-218: "Many of the most frequently committed crimes such as break and enter non dwelling, steal from retail and malicious damage" -- could be rephrased somewhat, e.g., "Many of the most frequently committed crimes, such as break-and-enter, retail theft, and malicious damage..."

3. Line 309: "Th BCR based solely on the DCE" -- change "Th" the "The".

4. Line 331: "While there was no data to support the impact of BackTrack" -- change "was" to "were". Same comment on line 377.

Round 2

Reviewer 1 Report

Authors have addressed the pointed issues. Despite the inherent uncertainty in the study, due to the nature of the topic, the effort spent by authors made the new version of the manuscript more scientifically sound, and hence, in my opinion, suitable for publication. A proofreading is still necessary to correct typos.

Author Response

Re: ijerph-1797165

Thank you to the reviewers for the latest comments on our manuscript ijerph-1797165.  I provide details of our revisions to the manuscript and our responses to the referees as follows:

Reviewer 1

We have proof-read and edited the manuscript as requested.

Thanks again to the reviewers for their insightful comments which have improved this paper.

Best regards,

Anthony Shakeshaft (on behalf of all authors).

Reviewer 2 Report

Thank you for the changes made based on previous comments. I believe all changes are satisfactory. There is some minor editing that needs to be done throughout the paper before publication. 

Author Response

Re: ijerph-1797165

Thank you to the reviewers for the latest comments on our manuscript ijerph-1797165.  I provide details of our revisions to the manuscript and our responses to the referees as follows:

Reviewer 2

We have undertaken editing throughout the paper as requested.

Thanks again to the reviewers for their insightful comments which have improved this paper.

Best regards,

Anthony Shakeshaft (on behalf of all authors).

Reviewer 3 Report

I would like to thank the authors for addressing my initial questions. I believe the presentation of the methods and results is improved in this revised draft. However, I still find the basis for the analysis somewhat confusing. Specifically, on p2 of the revised draft, the authors state that "Thirty-four males aged between 15 and 18 years met the criteria to be considered participants of BackTrack." Later, on p3, they state that "the central projection underpinning all the variable costs and benefits comprises the number of participants in the program and their level of participation, with school, with further education and/or with employment. For the specified program investment period, program data accurately recorded participant status."

However, there seems to be no data presented on the participants and key characteristics which the benefit-cost analysis would be based upon, e.g., extent of participant involvement in the program, proportion of respondents who completed the program, proportion who completed only part of the program, etc. This relates to my initial concerns on how potential evaluation data were incorporated into the analysis. At the very least, some description of the participants (e.g., a descriptive Table 1 of participants' characteristics and key outcomes) would be beneficial and this information, I think, should tied to the benefit-cost analysis.

Author Response

Re: ijerph-1797165

Thank you to the reviewers for the latest comments on our manuscript ijerph-1797165.  I provide details of our revisions to the manuscript and our responses to the referees as follows:

Reviewer 3

We note Reviewer 3’s comment that: “At the very least, some description of the participants (e.g., a descriptive Table 1 of participants' characteristics and key outcomes) would be beneficial…”.  To avoid any uncertainty, we have deleted Section 3.1 (lines 132-136) and added only the information that is directly salient to this paper into Section 2.1.1 (lines 118-120).  Specifically, we have referred readers to an existing published paper that provides a description of participants’ characteristics rather than try to re-report them in this paper, which is unnecessary given the focus is on the economic evaluation.

Thanks again to the reviewers for their insightful comments which have improved this paper.

Best regards,

Anthony Shakeshaft (on behalf of all authors).